# Accelerated Long-Fragment Circular PCR for Genetic Manipulation of Plant Viruses in Unveiling Functional Genomics

**DOI:** 10.3390/v15122332

**Published:** 2023-11-28

**Authors:** A. Abdul Kader Jailani, Anirudha Chattopadhyay, Pradeep Kumar, Oinam Washington Singh, Sunil Kumar Mukherjee, Anirban Roy, Neeti Sanan-Mishra, Bikash Mandal

**Affiliations:** 1Advanced Centre for Plant Virology, Division of Plant Pathology, Indian Agricultural Research Institute, New Delhi 110012, India; anirudhbhu@sdau.edu.in (A.C.); pradeepcsa7416@gmail.com (P.K.); oinamw@gmail.com (O.W.S.); sunilmukherjeeudsc@gmail.com (S.K.M.); anirbanroy75@yahoo.com (A.R.); 2International Centre for Genetic Engineering and Biotechnology, New Delhi 110067, India; neeti@icgeb.res.in; 3Plant Pathology Department, University of Florida, North Florida Research and Education Centre, Quincy, FL 32351, USA; 4Pulses Research Station, Sardarkrushinagar Dantiwada Agricultural University, Sardarkrushinagar 385506, India

**Keywords:** PCR based cloning, CGMMV vector, VIGS, LC-PCR

## Abstract

Molecular cloning, a crucial prerequisite for engineering plasmid constructs intended for functional genomic studies, relies on successful restriction and ligation processes. However, the lack of unique restriction sites often hinders construct preparation, necessitating multiple modifications. Moreover, achieving the successful ligation of large plasmid constructs is frequently challenging. To address these limitations, we present a novel PCR strategy in this study, termed ‘long-fragment circular-efficient PCR’ (LC-PCR). This technique involves one or two rounds of PCR with an additional third-long primer that complements both ends of the newly synthesized strand of a plasmid construct. This results in self-circularization with a nick-gap in each newly formed strand. The LC-PCR technique was successfully employed to insert a partial sequence (210 nucleotides) of the phytoene desaturase gene from *Nicotiana benthamiana* and a full capsid protein gene (770 nucleotides) of a *begomovirus* (tomato leaf curl New Delhi virus) into a 16.4 kb infectious construct of a *tobamovirus*, cucumber green mottle mosaic virus (CGMMV), cloned in pCambia. This was done to develop the virus-induced gene silencing vector (VIGS) and an expression vector for a foreign protein in plants, respectively. Furthermore, the LC-PCR could be applied for the deletion of a large region (replicase enzyme) and the substitution of a single amino acid in the CGMMV genome. Various in planta assays of these constructs validate their biological functionality, highlighting the utility of the LC-PCR technique in deciphering plant-virus functional genomics. The LC-PCR is not only suitable for modifying plant viral genomes but also applicable to a wide range of plant, animal, and human gene engineering under in-vitro conditions. Additionally, the LC-PCR technique provides an alternative to expensive kits, enabling quick introduction of modifications in any part of the nucleotide within a couple of days. Thus, the LC-PCR proves to be a suitable ‘all in one’ technique for modifying large plasmid constructs through site-directed gene insertion, deletion, and mutation, eliminating the need for restriction and ligation.

## 1. Introduction

The exponential growth of omics has resulted in the generation of a vast amount of genomic information, particularly concerning plant viruses. Traditionally considered as simple genetic elements [1], viruses have now become valuable resources for various structural and functional biology studies. These studies encompass a wide range of applications, including the synthesis of nanomaterials [2,3], creation of genetic circuits [4], design of biosensors [5], production of industrial biologics and bio-pharmaceuticals [6], and identification of gene functions [7]. Consequently, viruses have evolved into attractive genetic toolkits for research in functional genomics and synthetic biology [8]. The cloning of virus genomes and subsequent engineering are indispensable steps required to achieve the aforementioned goals. Despite their small size, plant virus genomes are highly complex, containing elements that are challenging to manipulate using conventional biotechnological tools. While the development of infectious clones for plant virus genomes is achievable with existing technology, complications arise from the presence of pseudo and cryptic prokaryotic promoter-like elements in the virus genome. These elements can hinder culturing in *E. coli* due to the expression of toxic proteins, presenting a challenge for researchers seeking desired modifications [9,10,11].

Technological challenges such as targeted silent mutations [12,13] or the insertion of introns [14] can be employed to address these problems. However, even when an infectious clone is obtained, further engineering for valorization is often tedious and may lead to unexpected outcomes. Challenges include the scarcity of suitable restriction sites within the genome, the presence of specific dinucleotide repeats (e.g., CpG or UpA), vulnerability to frameshift mutations due to single nucleotide additions or deletions, and the stability of foreign genes within the viral genome. Several approaches, including restriction-ligation-based methods, recombinase-based strategies, and PCR-mediated gene/genome cloning, have been developed to address these issues. While restriction-ligation methods are time-consuming and limited by the availability of preferable restriction sites, recombinase-based strategies like Gibson assembly [15,16], Gateway^®^ (Invitrogen), and In-Fusion™ (Clontech) have their limitations, such as high cost and the inability to clone larger fragments (>2 kb) [17,18,19].

To address cost concerns, recombinase-free (RF), polymerase chain reaction (PCR)-based cloning techniques have been invented, allowing gene insertion within the vector backbone through a simple PCR protocol without the addition of extra sequences. Early techniques like circular polymerase extension cloning (CPEC) [20,21,22] suffered from low yields of open circular amplicons [23]. Successive techniques like inverse fusion PCR cloning (IFPC) [24] and exponential megapriming PCR (EMP) [25] improved upon CPEC but still faced challenges such as low yield and complexities in phosphorylation and ligation steps [26]. To overcome these challenges, newer cloning techniques utilize the endogenous recombination activity of *E. coli* strains, such as overlap extension PCR and recombination (OEPR) and manmade cohesive termini (MCT) cloning [27]. While these techniques have reported success in cloning inserts up to 6 kb within a 4 kb–8 kb backbone vector, they face limitations in amplifying large vector backbones (>8 kb) [23,26].

In recent years, the improved overlap extension PCR (IOE-PCR) [28] has been introduced, surpassing the OEPR and MCT cloning strategies. Although tested for amplification up to a 12 kb plasmid, its suitability for creating site-directed mutations and deletions in plasmid constructs has not been demonstrated. Additionally, various PCR-based strategies are available for creating deletion and substitution-based mutations within plasmid constructs. Notably, the introduction of improved overlap extension PCR (IOE-PCR) [28] has surpassed the OEPR and MCT cloning strategies. Although tested for amplification up to a 12 kb plasmid, its suitability for creating site-directed mutations and deletions in plasmid constructs has not been demonstrated. Alongside methods for inserting fragments into plasmid constructs, various PCR-based strategies are now available for creating deletion and substitution-based mutations within plasmid constructs. Initially, quick-change was introduced with altered nucleotides in the middle of the primers; however, its self-complementarity led to decreased yields [29]. Subsequent strategies, such as ligation of fragment ends after PCR (LFEAP) mutagenesis [30], have been developed, either involving multiple PCR steps or requiring PCR purification and T4 DNA polymerase and polynucleotide kinase treatments. Although the LFEAP is claimed to be efficient for the mutagenesis of 15 kb plasmid constructs, the functionality of manipulated plasmid constructs remains unclear.

The extensive body of literature reflects the ongoing quest by researchers for highly specific, cost-effective, time-efficient, and less labour-intensive techniques for modifying plasmid constructs, especially those applicable to viral genome manipulation. Despite this, there is currently no ‘all in one’ technique available that encompasses site-directed gene insertion, mutation, and deletion for large plasmid constructs. In this study, we introduce a novel method called long-fragment circular-efficient PCR (LC-PCR)—a rapid, restriction, and ligation-free overlapping circular PCR. This technique involves providing an additional primer for the self-recircularization of the PCR molecule during the overlapping PCR step. We utilized the LC-PCR for the genetic manipulation (insertion, deletion, and mutation) of an infectious clone of cucumber green mottle mosaic virus (CGMMV). The CGMMV, a member of the tobamovirus family, infects a wide range of cucurbitaceous hosts, causing mild symptoms. Its genome is currently favored for constructing a suitable virus vector system for plant functional genomics studies [31], CGMMV transmission studies, as well as the expression of foreign proteins in cucurbits [32,33]. Emerging cucurbit-infecting viruses, such as the Squash vein yellowing virus (SqVYV), Cucurbit yellow stunting disorder virus (CYSDV), Cucurbit chlorotic yellows virus (CCYV), watermelon crinkle leaf-associated virus 1 (WCLaV1), and watermelon crinkle leaf-associated virus 2 (WCLaV2), the widespread impact of these viruses on cucurbit crops globally poses a significant challenge. However, the development of infectious clones for these viruses is crucial for gaining insights into their molecular mechanisms, replication, and pathogenesis. These clones serve as indispensable tools for exploring virus-host interactions, assessing disease resistance, and formulating targeted control strategies. Ultimately, they play a pivotal role in safeguarding cucurbit crops against the emerging threats posed by these viruses [34,35,36,37]. Despite their importance, the genome manipulation of these viruses encounters hurdles, primarily due to the presence of overlapping open reading frames (ORFs) in the genome. Consequently, the in planta functional efficiency of the CGMMV-based vector systems remains in question. To surmount these challenges, we employed the LC-PCR to introduce desired genetic changes. Subsequently, we conducted tests to evaluate the functionality of manipulated constructs, thereby validating the specificity and effectiveness of the LC-PCR method.

## 2. Materials and Methods

### 2.1. Plasmid Construct, Target Gene, and Test Plant

The full-length infectious cDNA clones of CGMMV and CGMMV-∆CP were obtained from pUCGMT7 and pCG6.0CG∆CP [32], respectively, through full-genome amplification and cloning into pBM1, a modified binary vector based on pCambia2300 (see Appendix A). These resulting constructs were designated as pBP4 and pBP4-∆CP, respectively, and were utilized for subsequent genome engineering processes. To acquire the partial sequence of the phytoene desaturase (*NbPDS*) gene, RT-PCR was performed using mRNA extracted from healthy *Nicotiana benthamiana*. Additionally, the full capsid protein gene (770 bp) of Tomato leaf curl New Delhi virus (ToLCNDV) was amplified from the ToLCNDV monomer via conventional PCR. Both amplicons were cloned into TA vectors (TA-NbPDS and TA-CP_ToLCNDV_) and sequenced for further use. All biological assays were conducted using a healthy *N. benthamiana* model plant in a controlled environment with a temperature range of 25–28 °C. The plants were subjected to a light cycle of 16 h and a dark period of 8 h.

### 2.2. Designing of Primers for Gene Insertion, Deletion, and Mutation 

The successful genetic manipulation (insertion, deletion, and mutation) of the CGMMV genome-based plasmid constructs (pBP4 and pBP4-∆CP) hinged on the careful design of primers. Typically, long primers with a high GC content (approximately 50–60%) and an annealing temperature ranging between 65–72 °C were crucial. For the targeted insertion of genes (*NbPDS* gene and CP_ToLCNDV_ gene) at specific positions within the CGMMV genome, long PCR primers were crafted by incorporating sequences from the target genes and the plasmid backbone where the insert was to be integrated (refer to Table 1). In general, forward (P1-F/P1a-F) and reverse (P2-R/P2b-R) primers, each around 35–40 nucleotides long, contained 20-nucleotide sequences from the target gene (*NbPDS* or CP_ToLCNDV_) and 15–20 nucleotide-long overhanging ends that were complementary to the sequences on the insertion target sites of the plasmid backbone. The products amplified in the first-round PCR (228 nucleotide sequences of the *NbPDS* gene) were then used as mega-primers (P3/P3c) for the second-round PCR. Notably, an additional reverse primer (P4/P4d) with a complementary sequence partly to the insert and the target plasmid was designed for the self-circularization of the amplicons (refer to Figure 1a–c). The primer set, P1-F, P2-R, and P4, was employed for inserting the *NbPDS* gene into the pBP4 plasmid (infectious clone of CGMMV). Similarly, another primer set, P1a-F, P2b-R, and P4c, was used for inserting the CP_ToLCNDV_ gene into the pBP4-∆CP, the CP-deficient pBP4, through LC-PCR.

To achieve the deletion of a large region in the CGMMV genome (pBP4), a set of three primers (P5-F, P6-R, P7-R) was meticulously designed (refer to Table 1). Specifically, for the targeted deletion of the sequence encompassing the helicase, RdRp domain of the replicase enzyme, and a portion of the MP ORF in the CGMMV genome, the forward primer (P5-F) was designed to bind specifically to the complementary sequence of the 5’ flanking region from nucleotide 834 to 869. Conversely, the reverse primer (P6-R) was designed to anneal at nucleotides 5272 to 5297. An additional reverse primer, P7-R, was created by retaining parts of the sequences from P5 and P6 primers, facilitating the self-circularization of the LC-PCR amplicons.

For introducing mutations in the CGMMV construct, primers (P-8, P-9, and P-10) were designed with altered tri-nucleotides at their overhanging ends to bring about desired mutations (specifically, CTG to TAC) within the CGMMV genome. An additional primer (P-10), designed by combining parts of P8 and P9, was included to aid in the self-circularization process. All primer sequences (refer to Table 1) were synthesized by GCC Biotech, India. These primers were instrumental for insertion (gene silencing and expression inserts), deletion, and mutation within the CGMMV genome.

### 2.3. LC-PCR Protocol for Gene Insertion 

To insert a gene, two rounds of PCR were conducted. In the first round, the target genes (*NbPDS* and CP_ToLCNDV_) were amplified from their template constructs (TA-NbPDS and TA-CP_ToLCNDV_) using different primer sets, namely P1-F, P2-R, and P1a-F, P2b-R, respectively. The primary PCR was carried out in a 25.0 µL reaction mixture, including 5.0 µL of 5× reaction buffer, 2.0 µL of 2.5 mM dNTP mix, 1.0 µL each of the forward and reverse primers (1.0 micromoles), 1.0 U of Phusion Taq polymerase (NEB, Massachusetts, USA), and DNase-free water to make up the volume. The thermocycling program included denaturation at 98 °C for 60 s, followed by 35 cycles of final denaturation at 98 °C for 10 s, primer annealing at 52 °C for 30 s, and synthesis at 72 °C for 30 s/kb. This was followed by one cycle of final extension at 72 °C for 10 min in a thermocycler (Verti, Invitrogen, Carlsbad, USA). The PCR amplicons from the primary PCR were gel-purified using the Wizard Gel and PCR Clean-up System (Promega, Madison, WI, USA). The purified PCR product from the primary PCR was then used as the mega primer (P3/P3c) in the second round of PCR, along with an additional primer (P4/P4d) required for amplification and self-circularization. The pBP4 and pBP4-∆CP plasmids were used as templates for inserting the *NbPDS* gene and CP_ToLCNDV_ gene, respectively.

In the second round of PCR, a 50.0 µL reaction mixture was prepared, including 10.0 µL of 5× reaction buffer, 4.0 µL of 2.5 mM dNTP mix, 100–150 ng of purified primary PCR product (P3/P3c), 2.0 µL of P4/P4c, 50.0 ng of template, 2.0 U of Phusion Taq polymerase (2.0 U/µL) (NEB, USA), and DNase-free water to make up the volume (50 µL) of the reaction mixture. The PCR program involved denaturation at 98 °C for 60 s, followed by 20 cycles. Each cycle included denaturing at 98 °C for 10 s, primer annealing, and synthesis at 72 °C for 50 s/kb, followed by one cycle of final extension at 72 °C for 10 min. For the removal of the original plasmid, the second-round PCR product (50.0 µL) was immediately treated with 2.0 µL of DpnI enzyme (10 U/µL) (Thermo Scientific) at 37 °C for 2 h, followed by enzyme inactivation at 80 °C for 20 min. The DpnI-treated PCR products (5, 7.5, and 10 µL) were directly used to transform 100 µL of chemically prepared competent cells (efficiency 10^5^ to 10^6^ cfu/mL) of *E. coli* JM109. The presence of *NbPDS* and CP_ToLCNDV_ genes in the recombinant colonies was confirmed through colony PCR using gene-specific primers (BM-556F and BM-910R for *NbPDS* gene, BM-1017F and BM-1018R for CP_ToLCNDV_ gene). Confirmation was also achieved through Sanger-based sequencing using the BM-489R primer, which is specifically bound to the 3′ UTR of CGMMV (see Appendix A).

### 2.4. LC-PCR Protocol for Deletion

To achieve a large deletion, a single round of the LC-PCR was performed using only two long primers (P5-R and P6-F), as described previously. A 16.4 kb large pBP4 construct (cDNA infectious clone of CGMMV) served as the template. The PCR was carried out in a 50 µL reaction mixture containing 10 µL of 5× reaction buffer, 4.0 µL of 2.5 mM dNTP mix, 1.0 µL each of P5-R and P6-F primers, 50 ng template plasmid (pBP4), 2.0 U of Phusion Taq polymerase (2.0 U/µL) (NEB, Massachusetts, USA), and DNase-free water for volume makeup. The PCR program included an initial denaturation at 98 °C for 60 s, followed by 10 cycles. Each cycle consisted of denaturation at 98 °C for 10 s, primer annealing and synthesis at 72 °C for 50 s/kb, and a hold at 25 °C for 60 s. After the completion of the 10th cycle, 2.0 µL of P7-F primer was added, and another 10 cycles of the above PCR program were continued. This was followed by one cycle of final extension at 72 °C for 10 min. Upon completion of the PCR program, DpnI enzyme treatment and subsequent transformation into *E. coli* JM109 competent cells were carried out, as described in the previous section. The clone was confirmed by colony PCR for the deletion of the plasmid using CGMMV UTR-specific primers (BM-486F and BM-489R) (see Appendix A) and subsequently by Sanger-based gene sequencing.

### 2.5. LC-PCR Protocol for Substitution-Based Mutation

Similar to the deletion strategy, a substitution-based mutation was carried out using a single round of LC-PCR. The target for mutation in the CGMMV genome was the triple base pair (nt) located at positions 1066–1068. For this LC-PCR protocol, long primers (P8-F, P9-R, and P10-R) were designed, incorporating the desired nucleotide at the overlapping ends. The preparation of the PCR mixture and the programming of the PCR protocol followed the same procedure as described for deletion-based LC-PCR. After the completion of the LC-PCR, the product was immediately treated with DpnI and then used to transform *E. coli* JM109 competent cells. The success of the site-directed mutagenesis was confirmed based on the sequence of the resulting mutant of pBP4 (refer to Appendix A).

### 2.6. Agro Infiltration of the Constructs in Plant

All engineered binary plasmid constructs were introduced into the GV3101 strain of *Agrobacterium tumefaciens* using the freeze-thaw method [38], employing appropriate antibiotic screening (kanamycin and rifamycin). The transformed colonies containing recombinant clones were further verified by colony PCR using gene-specific primers. The confirmed agro-cultures were subsequently used for syringe-infiltration on the abaxial leaf surface of 3–4-week-old *N. benthamiana* plants, along with a buffer containing 10 mM 2-(N-morpholino) ethane sulfonic acid (MES, pH 5.7), 10 mM MgCl_2_, and 150 µM Acetosyringone [39]. The infiltrated plants were then grown in an environmentally controlled room.

### 2.7. In Planta Functional Assay 

#### 2.7.1. Detection of Virus Infectivity in the Plant by CGMMV Genome-Based Constructs 

The emergence of typical mild mottle symptoms associated with CGMMV infection was regularly monitored up to 60 days post-infiltration (dpi) of the infectious clone. To assess the formation of virus particles in *N. benthamiana* plants inoculated with different binary constructs (refer to Appendix A), leaf extracts were placed on grids and stained with 2% uranyl acetate (pH 4.2). The grids were examined using a transmission electron microscope (TEM, JEOL JEM-1011, Tokyo, Japan).

Leaf samples, whether symptomatic or asymptomatic, were harvested from all inoculated plants at various time intervals to confirm the presence of the virus, predict gene silencing, and assess recombinant gene expression in the plant system through RT-PCR using gene-specific primers. Total plant RNA was isolated using a plant total RNA isolation kit (Promega, Wisconsin, USA), and cDNA was synthesized using reverse transcriptase (Superscript III, Invitrogen, Carlsbad, USA) enzyme with CGMMV 3′ terminal primer (BM489R) following the manufacturer’s protocol (see Appendix A). The PCR reaction mixture and program were conditioned as per the PCR protocol described above.

#### 2.7.2. Determination of the Host Gene-Silencing and Associated Phenotypic Changes

For *NbPDS* gene silencing, the observation of photobleaching symptoms was documented at 7-day intervals, starting from their initiation. The effectiveness of the gene silencing vector was assessed based on the intensity of photobleaching symptoms.

To quantify PDS transcript levels in the PDS-silenced plants, semi-quantitative RT-PCR was performed using total leaf RNA. RNA extraction from PDS-silenced and virus-infected plant leaves was carried out using a plant total RNA extraction kit (Promega, Madison, WI, USA). Total RNAs were treated with RNase-free on-column DNaseI. First-strand cDNAs were synthesized with an oligo(dT)20 primer using Superscript III reverse transcriptase (Invitrogen, Waltham, MA, USA). The infectivity of the CGMMV genome-based VIGS vector in plants was tested by RT-PCR using CGMMV CP gene-specific primers (BM 556F and BM 557R). Furthermore, the production of NbPDS RNA transcripts was detected based on the amplification of fragments containing CGMMV CP fused with NbPDS by BM-556F and BM-910R primers (see Appendix A) and fragments containing PDS fused with the 3′ UTR region of CGMMV using primer BM 909F and BM 489R (see Appendix A). Semi-quantitative RT-PCR was conducted to examine PDS mRNA transcript levels in silenced and healthy plant leaves in comparison to the housekeeping gene *NbActin 2* gene (used as the reference/standard). For the semi-quantitative RT-PCR assay, NbPDS gene-specific primers (BM909F and BM910R) were designed from the target region (see Appendix A) of the corresponding *NbPDS* gene selected for silencing.

#### 2.7.3. Analysis of the in Planta Heterologous Protein Expression (Coat Protein of ToLCNDV)

Healthy *N. benthamiana* plants were subjected to agro-infiltration with the pBP4-∆CP-CP_ToLCNDV_ construct and the ensuing phenotypic changes were monitored. Initially, the transcript of the CP gene of ToLCNDV expressed within the infiltrated leaf tissue of *N. benthamiana* plants was detected by RT-PCR using primers BM-1017F and BM-1018R (refer to Appendix A). Subsequently, the expression of CP_ToLCNDV_ in the infiltrated leaves was analyzed through SDS-PAGE. The plant infiltrated with only PBS buffer (mock plants) and the healthy plants served as controls. Leaf tissues (0.8 g) were ground to a fine powder in liquid nitrogen and then resuspended in 300 µL of protein extraction buffer containing 0.1 M KH_2_PO_4_, pH 6.5, 0.5 mM PMSF, and 10 mM β-mercaptoethanol. Total protein was resolved by 10% SDS-PAGE, and the gel was stained with Coomassie Brilliant Blue R250 [40].

#### 2.7.4. Evaluation of the Role of Helicase and RNA Dependent RNA Polymerase Domain of CGMMV in Genome Replication

To investigate the functional role of the helicase and RNA-dependent RNA polymerase (RdRp) domain in the genome replication of CGMMV, the pBP4-∆Rep construct was introduced via agro-infiltration into *N. benthamiana* plants. Phenotypic changes were monitored, and the replication of CGMMV was analyzed by RT-PCR. Samples from both infiltrated leaves and systemic leaves were collected at different time intervals (7 dpi, 14 dpi), and subsequently, cDNA was prepared. RT-PCR was conducted using CGMMV genome terminal primers (BM-486F and BM-489R) (see Appendix A).

#### 2.7.5. Assessment of the Infectivity of the CGMMV Mutant

To assess the impact of the mutation in the CGMMV genome (pBP4-Δ1066-68 construct), agro-infiltration was performed in *N. benthamiana* plants. The expression of symptoms in the infiltrated and systemic leaves of *N. benthamiana* was monitored up to 60 dpi in comparison with the wild-type CGMMV (pBP4 construct). Electron microscopy was conducted to examine the formation of virions of the mutant in the infected tissues. Furthermore, its replication was detected in systemic leaf tissues by RT-PCR using CGMMV CP gene-specific primers (BM 556F and BM 557R) (refer to Appendix A).

## 3. Results

### 3.1. Construction of CGMMV Genome-Based VIGS Vector and Its Validation through Silencing of NbPDS Gene 

The LC-PCR cloning technique was developed for the rapid and site-specific incorporation of the *NbPDS* gene and CP_ToLCNVD_ gene into CGMMV genome-based vectors (see Figure 1a). It entailed restriction and ligation-free insertion of the foreign genes into the target-specific site. In the initial PCR step, the *NbPDS* gene was amplified (Figure 1a) using primers P1-F and P2-R, which harbored overhang sequences homologous to the flanking regions of the insertion site situated at the end of CP ORF and the beginning of 3′UTR in the full-length CGMMV genome-based infectious constructs (pBP4). The PCR resulted in the amplification of a 0.21 kb fragment (see Figure 2b). The purified amplicons from the first-round PCR were utilized as long primers (P3) for the second-round PCR, binding to their complementary sequences of the pBP4 plasmid specifically at the site of insertion. An additional primer, P4, containing the complementary sequences to the P1-F primer at its 5′ end and to the pBP4 plasmid at its 3ꞌ end, was utilized in the same PCR reaction for the self-circularization of the amplicons of 16.6 kb, integrating the insert into the vector (pBP4 + NbPDS) with nicked ends (see Figure 1a and Figure 2c). The circular parental plasmid (pBP4) was selectively digested with *DpnI* enzyme treatment, and the remaining non-covalently closed circular PCR molecules (the final product) were used to transform *E. coli* cells to seal the gaps (see Figure 2d). The colony PCR with CGMMV CP gene-specific forward primer (BM-556F) and NbPDS reverse primer (BM-910R) resulted in the amplification of a 0.70 kb fragment, indicating the successful insertion of PDS into the pBP4 plasmid (see Figure 2d). The LC-PCR and *E. coli* transformation processes were repeated five times, resulting in more than 90 colonies. Among them, 25 colonies were tested with a 96% success rate in colony PCR (see Table 2).

The efficacy of the CGMMV-based VIGS vector (pBP4-NbPDS) (refer to Figure 2a) was assessed through a planta gene silencing assay. The native *PDS* gene in *N. benthamiana* was targeted to visualize the silencing phenotype, characterized by chlorophyll photobleaching on the leaves. A partial fragment of the *NbPDS* gene, derived from the mRNA sequence, was inserted in the sense orientation into the CGMMV genome through the LC-PCR to create a VIGS vector. Following agro-infiltration of the VIGS vector, all six *N. benthamiana* plants began exhibiting the photobleaching phenotype from 16 dpi to 45 dpi (see Figure 2e–h). This outcome demonstrated the successful development of a CGMMV genome-based VIGS vector. The effectiveness of the VIGS vector was confirmed through RT-PCR, which was conducted to amplify the CP fused with NbPDS from the mRNA of the photobleached leaves. RT-PCR utilized CP forward primer (BM 556F) and PDS reverse primer (BM910R), as well as PDS forward primer (BM909F) and 3′ UTR reverse primer (BM 489R). The results exhibited the expected amplification of a 700 bp long CP fused with PDS and a 550 bp long PDS fused with UTR regions (see Figure 2i).

To investigate the down-regulation of *NbPDS* gene expression in the photobleached leaves, the transcript level of the *NbPDS* gene was assessed through semi-quantitative RT-qPCR, revealing reduced expression compared to the housekeeping gene (Actin2) in the photobleached leaf samples at 16 dpi (see Figure 2k). Healthy as well as CGMMV-infected leaf tissues were used as controls to detect NbPDS transcript levels. However, no significant change in the expression level of NbPDS transcripts compared to Actin2 was observed in the healthy and CGMMV-infected leaf samples (see Figure 2j). The decrease in mRNA levels of the *NbPDS* gene, as well as the photobleaching phenotype on leaves, confirmed the post-transcriptional silencing efficiency of the CGMMV-based VIGS vector constructed through LC-PCR.

### 3.2. Heterologous Expression of CP_ToLCNDV_ Gene Using CGMMV-Based Protein Expression Vector

The strategy for gene insertion through the LC-PCR was employed to construct a CGMMV genome-based protein expression vector (pBP4-∆CP) (refer to Figure 3a). The 770 nt sequence of the CP_ToLCNDV_ gene was integrated into the pBP4-∆CP vector within the truncated CP coding frame, encompassing 105 nt of the 5′ end of CP as the subgenomic promoter. The insertion process is illustrated diagrammatically in Figure 1a. The CP_ToLCNDV_ gene (770 nt) was amplified (Figure 3b) using long primers (P1a-F and P2b-R) containing overlapping sequences complementary to the 3′ and 5′ flanking regions of the insertion site in the pBP4-∆CP vector. The purified PCR amplicons from the first round of PCR were used as mega primers (P3c) for the insertion of the CP_ToLCNDV_ gene within the CGMMV genome-based vector (pBP4-∆CP) using the LC-PCR protocol. For self-circularization, an additional primer, P4d, containing complementary sequences of the insert and vector, was added in the second round of PCR, generating non-covalently closed, circular, recombinant PCR amplicons of 16.8 Kb (Figure 3c). The LC-PCR was repeated 10 times, and following transformation, a total of 80 colonies were obtained. The colony PCR test of 25 colonies showed 22 containing the CP of ToLCNDV, indicating an 88% success rate (refer to Table 2).

*N. benthamiana* plants agro-infiltrated with the pBP4 construct (infectious clone) displayed symptoms of green mottle mosaic (Figure 3e). However, plants agro-infiltrated with pBP4-∆CP, containing a partial CP gene of CGMMV, and pBP4-∆CP-CP_ToLCNDV_ construct, containing the full-length sequence of the CP_ToLCNDV_ gene, did not exhibit any symptoms (Figure 3f,g). RT-PCR analysis with CP_ToLCNDV_ gene-specific primers revealed the amplification of the expected 770 bp band in the leaves of all *N. benthamiana* plants agroinfiltrated with the pBP4-∆CP-CP_ToLCNDV_ construct (Figure 3h). However, such amplification could not be obtained in plants agroinfiltrated with pBP4 and pBP4-∆CP. These results confirm the expression of mRNA transcripts of the CP gene of ToLCNDV from the pBP4-∆CP-CP_ToLCNDV_ constructs in the infiltrated leaves of *N. benthamiana*. Additionally, the expression of a 34.0 kDa protein from this transcript was detected at 5–15 dpi in SDS-PAGE electrophoresis (Figure 3i).

### 3.3. Deletion of Helicase and RNA Dependent RNA Polymerase Restricts the Self-Replication of Deconstructed CGMMV Genome

An internal deletion of a 4.4 kb genomic segment (836–5271 nt), encompassing the helicase domain, RNA-dependent RNA polymerase domain of the replicase enzyme, and the sub-genomic promoter region of the movement protein ORF in the CGMMV genome, was generated in the pBP4 (CGMMV infectious clone) construct using the LC-PCR protocol (Figure 1b). For the deletion of the specific region of the plasmid construct, the first-round PCR, utilizing the primer set P5-R and P6-F, which specifically annealed to both ends being deleted in the CGMMV genome (Figure 4a), resulted in the amplification of a deconstructed genome fragment of approximately 12.0 kb (Figure 4c). The addition of the P7-F primer facilitated the self-circularization of the deconstructed fragments into non-covalently closed, circular PCR molecules. Following DpnI treatment of the final PCR product and E. coli transformation, a total of 150 colonies were obtained in three independent LC-PCR experiments. Testing 25 randomly selected colonies with PCR using terminal primers (BM-486F, BM-489R) showed the amplification of a 2.0 kb fragment, representing the deconstructed genome of CGMMV in all 25 colonies, indicating a 100% success rate (Table 2, Figure 4d). The recombinant colonies were further confirmed through restriction digestion with terminal restriction sites, BamH1 and XbaI, resulting in the release of the 2.0 kb fragment from the pBP4-∆Rep construct (Figure 4e), while the parental pBP4 construct was used as the negative control, releasing 6.4 kb fragments representing the full CGMMV genome upon digestion with the same enzymes (Figure 4f). These analyses demonstrate the successful creation of a deconstructed CGMMV genome construct, pBP4-∆Rep, by deleting a large internal sequence (835–5271 nt) of the pBP4 clone.

The functionality of the deconstructed genome was assessed by agro-infiltrating the pBP4-∆Rep construct into *N. benthamiana*. In contrast to the pBP4 construct, which self-replicated and induced typical green mottle mosaic symptoms in *N. benthamiana* (Figure 4i), the pBP4-∆Rep replicon displayed an inability to self-replicate, as indicated by RT-PCR analysis within infiltrated tissue and did not elicit any symptoms in either the inoculated or systemic leaves (Figure 4g).

### 3.4. Site-Directed Mutagenesis Leads to the Amino Acid Substitution in the CGMMV Genome and Subsequent the Abolition of Virus Symptoms

The LC-PCR protocol was employed to introduce a site-directed mutation into the pBP4 clone (Figure 1c). The CTG trinucleotides positioned at the 1066–1068th nt in the CGMMV genome were targeted for substitution with TAC trinucleotides, which were incorporated at the overhanging ends of both overlapping primers. The P-8 primer selectively annealed at 1065 to 1092 nt, and the reverse primer (P-9) annealed at 1069 to 1100 nt in the pBP4 construct (Figure 5a). An additional primer, P-10, was used for the self-circularization of single-base substituted long circular PCR molecules (16.4 kb) from the pBP4 clone (Figure 5d); the resulting amplicons were designated as pBP4-Δ1066-68. After DpnI treatment, the transformation of *E. coli* with the pBP4-Δ1066-68 plasmid resulted in 200 colonies in five independent experiments. PCR testing of randomly selected 25 colonies showed a 100% positive rate (Table 2). Furthermore, the sequencing of five PCR-positive clones demonstrated the replacement of CTG (leucine) with TAC (tyrosine) in all the clones (Figure 5c). The results showed a high success rate of creating site-directed mutation by LC-PCR.

The tri-nucleotide change from CTG to TAC resulted in a single amino acid substitution from leucine (L) to tyrosine (Y) in the helicase domain of CGMMV. *N. benthamiana* plants agro-infiltrated with the pBP4-Δ1066-68 construct did not exhibit symptoms of CGMMV. In contrast, all plants agroinfiltrated with the pBP4 construct displayed the typical green mottle mosaic symptoms of CGMMV (Figure 5e). Interestingly, electron microscopy revealed the presence of many rod-shaped virions (300 × 18 nm) in plants infiltrated with the mutated construct (pBP4-Δ1066-68) (Figure 5g). A comparison with pBP4-infiltrated plants showed a similar number of virions in plants infected with the mutated construct. Furthermore, RT-PCR using CGMMV CP gene-specific primers (BM 556F and BM 557R) showed the amplification of a 480 bp CP gene fragment in both the inoculated and systemic leaves of *N. benthamiana* plants infiltrated with the mutated construct (Figure 5h). The mutation resulted in latent infection. These results demonstrate that the LC-PCR is highly efficient in creating mutations for studying functional genomics in CGMMV.

## 4. Discussion

Over the past decade, next-generation sequencing has led to the accumulation of a vast amount of genomic information. However, the decoding of many complex genomes still lags, underscoring the need to prioritize the exploration of the functional identity of genome sequences. Various strategies, such as gene silencing, ectopic expression, and subcellular protein localization, are commonly employed to elucidate gene structure and functions. An alternative method for producing valuable proteins directly within the host organism involves the use of virus-based expression vectors. Plant virus-derived expression systems [41,42] offer several advantages, including increased expression levels, rapid production, scalability, safety, and cost-effectiveness. Defining functionality relies on the efficient and specific cloning of target genes into specialized vectors and the alteration of sequences in the gene construct.

Traditionally, the restriction-ligation-based cloning approach has been the ‘cost-effective’ option for the majority of researchers, despite being tedious, time-consuming, and inefficacious, especially when handling large plasmid constructs. Many cloning vectors for plant functional genomics studies contain multiple cloning sites (MCS) within their targeted gene expression cassettes, including 35S promoters and nos terminators. However, the cloning process is often restricted by a lack of suitable restriction sites. Even if successful, cloning can introduce undesirable extra sequences that affect the configuration of the expressed protein, particularly in studies of plant-virus interactions where the construct contains the full genome with multiple genes and regulatory elements. As a result, researchers are exploring new methodologies that make gene/genome manipulation more accessible in terms of time and cost, while maintaining a high success rate.

With the advent of PCR technology, the outmoded practice of restriction-ligation-based gene cloning directly from source DNA is being replaced. PCR-based strategies are now utilized to develop recombinase-free (RF) cloning techniques for the site-specific modification of plasmid constructs. Several techniques, initially synthesized based on the overlapping extension PCR site-directed mutagenesis protocol, have been invented. The LC-PCR is one such unique technique developed for the genetic manipulation of plant virus genome-based vector constructs simply and efficiently. In the present study, the insertion of foreign genes like *NbPDS* within the CGMMV genome-based vector was performed to establish its utility as a VIGS vector. A large deletion was created within the replicate domain of the CGMMV genome to generate a replication-deficient deconstructed genome using the same strategy. Additionally, a few nucleotide changes were made to create new mutants for functional validation.

This technique was employed to modify CGMMV-based plasmid constructs through insertion, deletion, and mutation. Previously, similar techniques like modified Quick-Change^TM^ [29], modified overlapping extension PCR [27], and Improved Overlapping Extension PCR [28] were developed for site-directed deletion, insertion, and substitution mutagenesis of plasmid constructs. Methodologies were refined over time to achieve better success rates. In modified Quick-Change^TM^, the PCR steps were prolonged, especially in the final extension stage, continuing for up to 30 min [29]. In contrast, a touchdown PCR strategy with a modified final extension step at 72 °C for 10 min was used to create multiple mutations in modified overlapping extension PCR [27]. Although seemingly improved, the strategy involving the restriction digestion of PCR products followed by ligation introduced additional complications.

Our LC-PCR strategy closely resembles the simple overlapping extension PCR (OEP), which is purported to be effective for cloning various genes of different sizes, such as GFP (1 kb), GusA (1.9 kb), lacZ (3.2 kb), etc. [43]. However, in this strategy, cloning efficiency often decreases with increasing insert length. To address this issue, we introduced an additional primer for each strategy (insertion, deletion, and base-substitution) in the LC-PCR protocol. This extra primer is crucial for self-circularization, akin to the overlap extension PCR and recombination in vivo (OEPR) cloning [26], manmade cohesive termini (MCT) [23], and improved overlapping extension PCR (IOEP) [28]. An extra reverse primer is essential for the exponential amplification of desirable products.

In OEPR and MCT cloning strategies, the insertion of relatively longer (1.0–6.0 kb) and multiple (2–4) fragments into the targeted sites of the vector backbone was made effective. In each case, the pGADT7 vector (7988 bp) was used to clone up to 6.0 kb inserts, though it was not examined for cloning a fragment larger than 6.0 kb. Additionally, the ability of these approaches for genetic manipulation, such as deletion and mutation, is not well-documented. On the other hand, the IOEP cloning strategy was tested for the insertion and replacement of multiple DNA fragments sized about 1.7 kb at multiple sites within the vector backbone. The novelty of this approach lies in the use of T4 DNA polymerase with 3′-5′ exonuclease activity in place of DpnI to enhance transformation efficiency [28]. While all these techniques claim efficiency for insertion, deletion, and substitution at multiple sites, experimental evidence regarding their utility for creating large plasmid constructs is still lacking. Furthermore, functional validation of these constructs has not been depicted.

Our LC-PCR cloning strategy expands the limits of handling the size of the plasmid for genetic manipulation beyond existing PCR-based cloning techniques. The LC-PCR enables the insertion of gene fragments within a >16 kb vector, and it requires special care in designing primers that are highly specific to the targets and non-self-overlapping to avoid primer dimer formation. Standardizing the PCR protocol parameters is also essential for obtaining better yields when modifying larger plasmid constructs. In our experimentation, all these modifications were successfully carried out in a large-sized (>16 kb) plasmid construct containing the CGMMV genome within a modified pCambia-2300 backbone. In practical terms, PCR-based cloning or modification of large vectors (>15 kb) is very challenging and has not been reported yet. Notably, the majority of plant binary vectors (pBin, pGreen, pCambia series) are larger [44], making them cumbersome to clone/modify through PCR-based strategies. Therefore, our LC-PCR provides a unique option for constructing and modifying large-sized plasmid backbones and can serve as an alternative to the recombinase-based, highly expensive ‘Gateway’ system and ‘In fusion’ technology for the rapid generation of various constructs, especially for functional genomics [45]. The lack of dependency on highly expensive enzymes makes our technology effective for large-scale cloning in most small laboratories.

In this context, we have showcased the adaptability of our technique by crafting plasmid constructs for functional genomics studies. Leveraging this technology, we’ve manipulated a plant virus genome through gene insertion, deletion, and mutation approaches. Furthermore, we’ve substantiated the functional validation of these constructs. Initially, this methodology was established for the development of a CGMMV-genome-based VIGS vector and a protein expression vector through the insertion of foreign genes, namely *NbPDS* and the CP gene of ToLCNDV. Subsequently, the functionality of these constructs was confirmed in *N. benthamiana* through successful gene silencing and the transient expression of foreign proteins.

The CGMMV-based VIGS vector, containing the *NbPDS* gene at the CP end in the sense orientation, induces photo-bleaching of leaves after 16 dpi with very mild disease symptoms; prominent phenotypic changes (photobleaching) are recorded up to 60 dpi. Our CGMMV-based VIGS vectors are highly effective in gene silencing in *N. benthamiana*. Previously, the CGMMV full-length genome-based VIGS vector was developed by [31] using a sense and hairpin strategy. Initially, they inserted the PDS gene into the *Hind*III restriction site located at the end of the CP ORF of the CGMMV genome, but no photobleaching was reported in watermelon, despite virus infection. This might be attributed to the instability of foreign sequences within the *Hind*III restriction site [32]. To address this issue, they introduced numerous restriction-ligation-based genome modifications, such as the duplication of the CP sub-genomic promoter with the BamH1 restriction site, for the construction of the VIGS vector. They achieved successful gene silencing in various cucurbitaceous hosts. However, their claim regarding the non-host, model plant *N. benthamiana*, is questionable due to the lack of appropriate phenotypic evidence.

Here, we report the first ‘successful’ evidence of the CGMMV-based VIGS vector for the silencing of plant endogenous genes in a model plant. The *NbPDS* gene was inserted just after the stop codon (within the *Hind*III site) of the CP-ORF through LC-PCR, rather than a restriction-ligation-based strategy and typical photobleaching symptoms were recorded in *N. benthamiana* starting from 16 dpi to 60 dpi. This finding establishes the applicability of the LC-PCR for the rapid construction of CGMMV-based VIGS vectors, without extensive genome modification. Furthermore, RT-PCR-based detection reveals the stabilization of the PDS insert within the recombinant virus even after 45 dpi, simultaneously resolving the problem of the large insert being expelled from the *HindI*II restriction site of the recombinant CGMMV-based vectors after 18 dpi [32]. Thus, the LC-PCR is also found to be suitable for the development of stable recombinants for functional assays.

Additionally, this strategy is employed for the insertion of a larger size gene (770 bp long CP gene of ToLCNDV) into the CGMMV genome-based infectious cDNA construct. The subsequent expression of the 34 kDa fused coat protein of ToLCNDV in *N. benthamiana* proves the functionality of the recombinant construct. To our knowledge, this is the first report of planta expression of the coat protein of ToLCNDV. Previously, the expression and purification of the 30 kDa coat protein of Tomato yellow leaf curl virus (ToYLCV), another member of the genus Begomovirus, were reported in *E. coli* [46] and the Baculovirus [47] expression system. We also attempted to express the coat protein of ToLCNDV under the pET28a backbone in the *E. coli* BL21 DE3 strain but were unable to obtain expressed CP protein. Therefore, we shifted to the in planta expression strategy and successfully achieved its transient expression in the plant using the CGMMV-based protein expression vector system. This provides valuable evidence to justify the potential of our technique.

Subsequently, the same methodology was employed to induce a substantial deletion (approximately 4.4 kb) within the CGMMV genome, removing a significant section of the replicase and movement protein (from 835 bp to 5271 bp) to impede its self-replicating ability. This sheds light on the functional significance of each domain of the replicase enzyme in the genome replication of CGMMV. Previously, similar deletion mutants were generated in other plant viruses, such as TMV, primarily through a restriction-ligation-based strategy. However, the major limitation lies in the unavailability of desirable restriction sites at specific positions in the genome. Therefore, we explored the use of a PCR-based strategy for creating deletion mutants of the plant virus to decipher its functional analysis. This one-round PCR strategy is also valuable for the rapid and site-specific elimination of gene(s) or gene segments from the genome for their functional annotation.

Likewise, the introduction of substitution mutagenesis within the infectious cDNA construct of CGMMV is feasible using the LC-PCR protocol. In this case, a single-base mutated overlapping primer pair with an additional reverse primer is sufficient to create a single-base substitution within a large plasmid construct (>16 kb). The tri-nucleotide change at the 1066-68th position (CTG-TAC) results in the amino acid substitution from Leucine (L) to Tyrosine (Y), although it has a minimal impact on the pathogenicity of CGMMV. Various overlapping PCR-based deletion and substitution mutagenesis strategies have been described earlier. The Quik-Change^TM^ Site-Directed Mutagenesis System (Stratagene, La Jolla, CA, USA) is a pioneer in this field [29]. However, self-annealing of completely complementary primers limits the PCR yield, often leading to many false positives [48]. Subsequently, several other strategies with enhanced efficiency and higher fidelity have evolved. Among them, LFEAP mutagenesis (Ligation of Fragment Ends After PCR) [30] and Improved Overlap Extension PCR (IOEP) [28] have emerged as leaders. However, these strategies involve a two-step PCR and depend on T4 DNA polymerase, making them a cumbersome and costly endeavor for creating various mutations in the plasmid. In our strategy, a simple, one-round PCR is sufficient for creating deletion, insertion, and substitution-based mutagenesis, allowing the creation of numerous mutants in less time and cost.

## Figures and Tables

**Figure 1 viruses-15-02332-f001:**
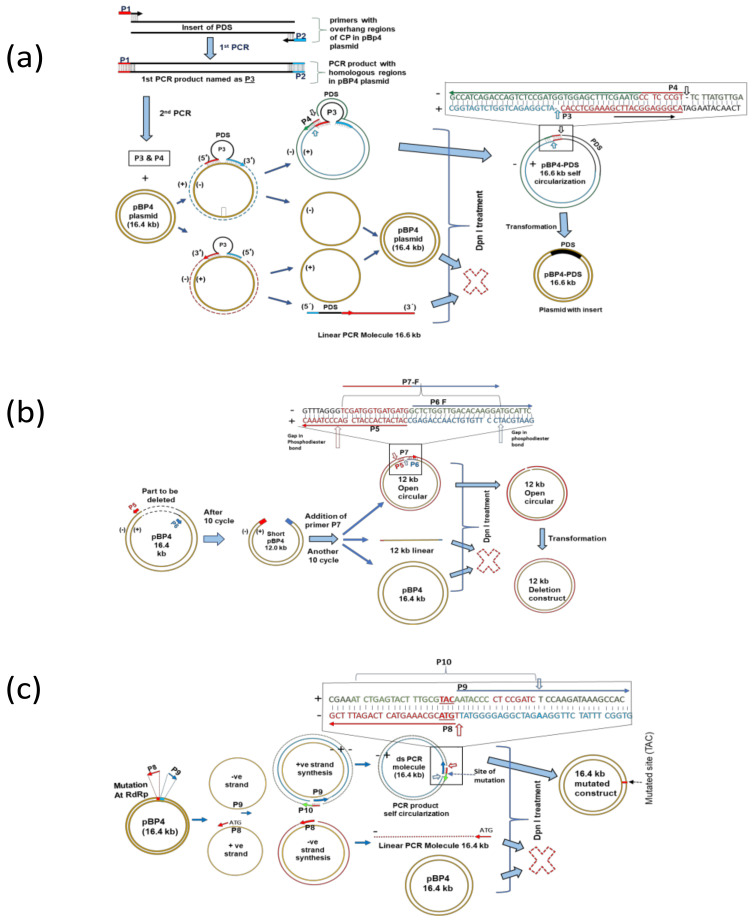
Schematic diagram of long-fragment circular-efficient PCR (LC-PCR) based cloning technique for insertion, deletion, and mutation in larger plasmid construct. (**a**) Insertion: In the 1st PCR, the insert was amplified using the forward primer (P1-F) and reverse primer (P2-R), each of which at the 5′ end contained the overlapping sequences of the target locations of insertion in the plasmid construct, pBP4. The amplified products were purified and used as the long primers (P3) in the 2nd PCR, where both ends of P3 annealed to the complementary sequences in the plasmid backbone. PCR was performed for the amplification of the plasmid containing the targeted insert. In the 2nd PCR, an additional reverse primer (P4) was used for the self-circularization of the amplicons, which was expected to yield a new recombinant circular molecule containing one gap/nick on the positive strand at the 5′ end of P4 primer and another gap/nick on the negative stand at the 5′ end of P1-F primer. (**b**) Deletion: The primers P5-R and P6-F were used for the specific annealing to the complementary sequences in the vector backbone to delete the desired region by exponential LC-PCR and followed by self-circularization by using primer, P7-F. (**c**) Mutation: the primer P8-F and P9-R contained triple mutated bases at their terminal ends and bound specifically to the vector backbone to generate the mutated PCR product, which was circularised by using a primer, P10-R. Removal of parent plasmid and transformation: For the selective removal of the parent plasmids, DpnI treatment was applied for their (parent plasmids) specific restriction digestion. Thereafter, the intact circular, but non-covalently closed the LC-PCR amplicons were transferred into the competent *E. coli* JM109 cells to seal the gaps and to convert the LC-PCR amplicons into the circular recombinant plasmid molecules.

**Figure 2 viruses-15-02332-f002:**
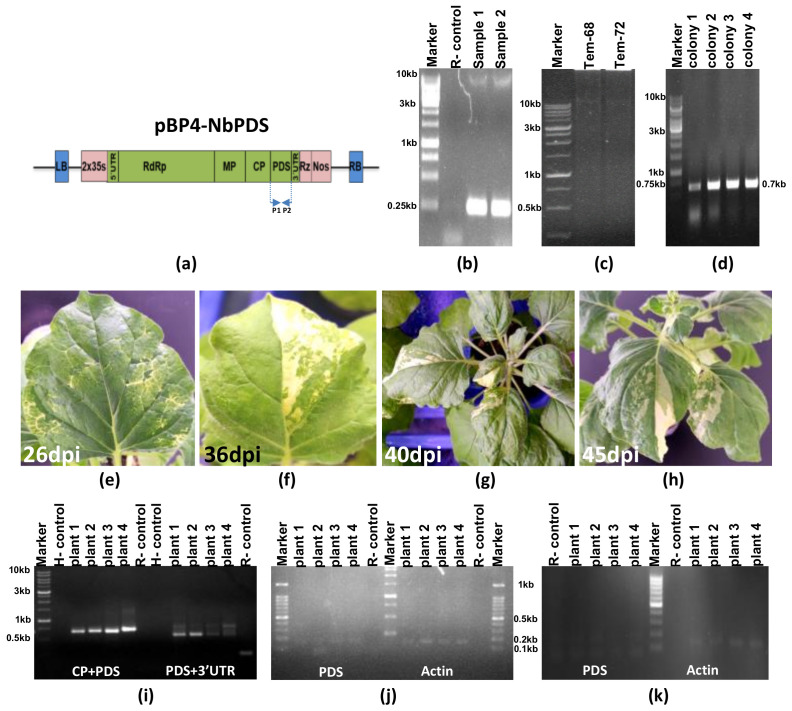
Insertion of the phytoene desaturase (*NbPDS*) gene segment from *N. benthamiana* into the CGMMV genome-based infectious construct (pBP4) through long-fragment circular-efficient PCR (LC-PCR) and its validation through in planta gene silencing. (**a**) The design of CGMMV-genome-based VIGS vector containing *NbPDS* gene, P1, and P2 shows the location of the inserted *NbPDS* gene at the end of the coat protein (CP) coding frame of CGMMV. (**b**) The PCR amplified *NbPDS* gene fragment (0.21 kb) using P1-F and P2-R primers in the 1st round PCR, (**c**) The LC-PCR amplified product (16.6 kb) containing the *NbPDS* gene inserted in pBP4 through 2nd round of LC-PCR. (**d**) The colony PCR-based conformation of CGMMV genome vector containing *NbPDS* gene using CGMMV CP gene-specific primer (BM-556F) and *NbPDS* reverse primer (BM-910R) yielding 0.70 kb fragment. (**e**–**h**) Development of photobleaching symptoms on *N. benthamiana* agro-infiltrated with pBP4 containing *NbPDS* gene at different days post inoculation demonstrating the successful silencing of PDS gene by the CGMMV-based gene silencing vector. (**i**) The RT-PCR amplification of the fragments containing the CGMMV CP fused with *NbPDS* by BM-556F and BM-910R primers and that of fragments containing PDS fused with 3′ UTR region of CGMMV using primer BM909F and BM 489R from the mRNA of the pBP4-PDS inoculated plant leaves. (**j**) Detection of the *NbPDS* and *NbActin* mRNA transcripts in pBP4 infected plants by semiquantitative RT-PCR using primer sets: BM909F and BM910R, and BM 849F and BM 859R, respectively. (**k**) Semiquantitative RT-PCR of *NbPDS* as well as *NbActin* gene fragments in the photobleached tissues of pBP4-PDS inoculated plants indicating the successful silencing of the PDS gene.

**Figure 3 viruses-15-02332-f003:**
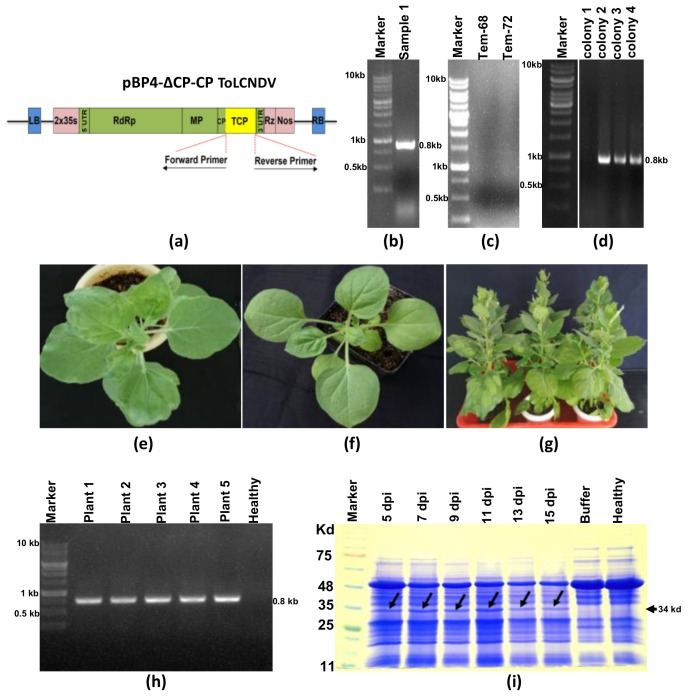
Insertion of ToLCNDV CP gene into the CGMMV genome (pBP4-∆CP-CP_ToLCNDV_) through long-fragment circular-efficient PCR (LC-PCR) and its in planta expression analysis. (**a**) The schematic diagram of the (pBP4-∆CP-CP_ToLCNDV_) construct indicates the site-specific insertion of the CP_ToLCNDV_ gene after the CGMMV CP-sub genomic promoter. (**b**) The agarose gel image showed the amplified fragment (770 bp) of the CP_ToLCNDV_ gene using P1a-F and P2b-R primers that contained the complementary sequence of CGMMV CP gene at its overhanging ends. (**c**) The gel image shows the insertion of the CP_ToLCNDV_ gene within the pBP4-∆CP vector through LC PCR using step-1 PCR product as a long primer P3c. (**d**) The gel picture shows the conformation of transformed *E. coli* colonies using CP_ToLCNDV_ gene-specific primers (BM-1017F, BM-1018R). (**e**) *Nicotiana benthamiana* plant showing typical green mottle mosaic symptoms following agro-infiltration of pBP4 construct (infectious clone of CGMMV). (**f**) *N. benthamiana* plant showing no symptoms following agro-infiltration of pBP4-∆CP (CGMMV construct containing partial CP of CGMMV). (**g**) *N. benthamiana* plant infiltrated with pBP4-∆CP-CP_ToLCNDV_ construct showing no symptoms of CGMMV. (**h**) RT-PCR showing detection of the CP gene transcript (770 nt) of ToLCNDV expressed in the leaf tissues of *N. benthamiana* plants infiltrated with the pBP4-∆CP-CP_ToLCNDV_ construct at 7 dpi using primers BM-1017F and BM-1018R. (**i**) SDS PAGE showing the expression of 29 kDa coat protein of ToLCNDV in *N. benthamiana* leaf tissues infiltrated with pBP4-∆CP-CP_ToLCNDV_ construct, but expression of CP_ToLCNDV_ in the mock-infiltrated (PBS-buffer) or non-infiltrated control plants (Healthy).

**Figure 4 viruses-15-02332-f004:**
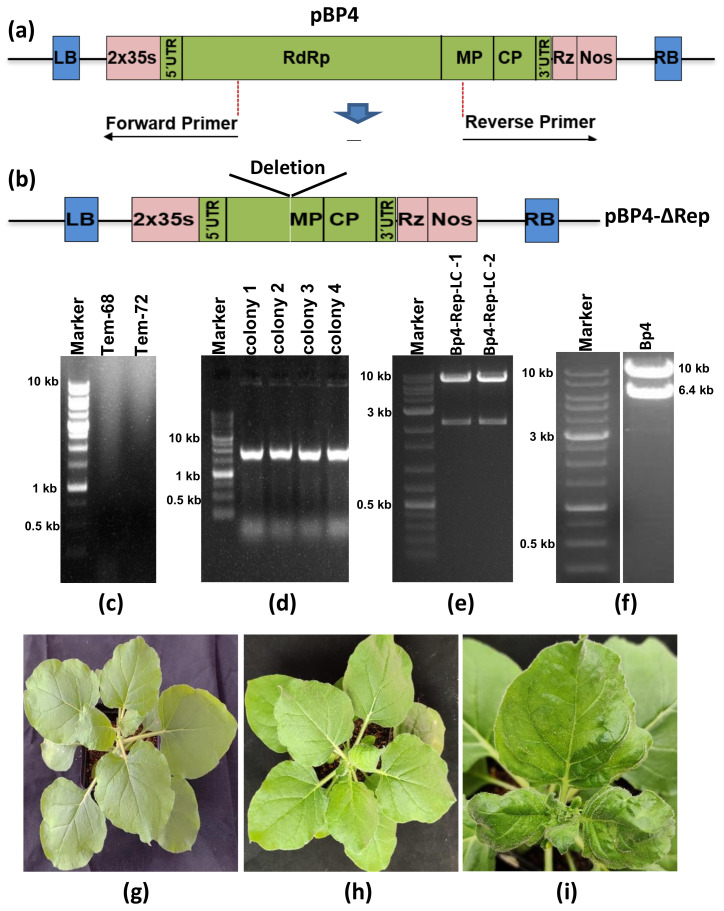
Site-specific deletion of the gene(s) or partial genome of CGMMV via long-fragment circular-efficient PCR (LC-PCR) to decipher the functional significance in viral replication studies. (**a**) A schematic diagram represents the strategy of site-specific deletion of the CGMMV genome from helicase to partial MP ORF. (**b**) The schematic diagram shows the deconstructed genome of CGMMV following the deletion of a large region. (**c**) Amplified fragment (12 kb) of partially deleted CGMMV genome-based plasmid construct through the LC-PCR at different annealing temperatures (68 °C, 72 °C). (**d**) The colony PCR-based conformation of the partially deconstructed CGMMV genome by using terminal primers (BM486F, BM489R) amplifying 2.0 kb fragment. (**e**) Separation of the 2.0 kb fragment from the pBM-1 vector backbone construct following the digestion with the BamH1 and XbaI restriction enzymes located at the terminal ends of CGMMV genome confirming the site-specific deletion of the gene(s) or genome via LC-PCR. (**f**) The restriction digestion of the full-length CGMMV genome construct releases a 6.4 kb fragment from the pBM-1 vector. (**g**) Absence of typical mottle mosaic symptoms on the pBP4-∆Rep inoculated *N. benthamiana* plant indicating inability of replication of deconstructed CGMMV genome due to the loss of helicase and RdRp domain of replicase. (**h**) The buffer inoculated plants as a negative control. (**i**) The pBP4 inoculated plants showed typical green mottle mosaic symptoms at 10 dpi. For further confirmation, RT-PCR was performed using the virus-specific terminal primers (BM-486F and BM-489R), but the pBP4-∆Rep replicon remained undetected within infiltrated tissue (not shown in the figure).

**Figure 5 viruses-15-02332-f005:**
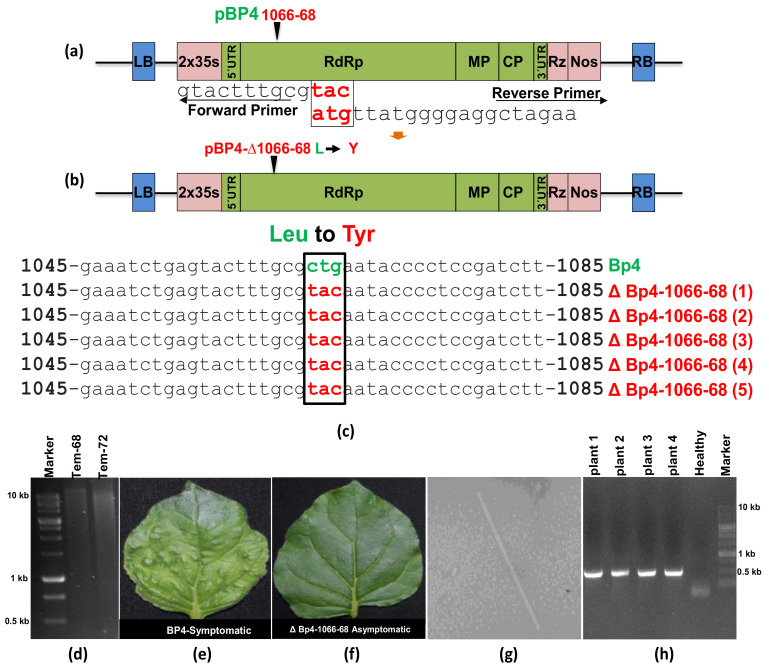
The site-specific mutation within the CGMMV genome through long-fragment circular-efficient PCR (LC-PCR). (**a**) The schematic diagram shows the strategy of introducing the site-specific mutation within the CGMMV genome through the LC-PCR using long primers (P8-F, and P9-R) with mutated nucleotides at their overlapping ends. (**b**) Shows the location of the mutation in the CGMMV genome, where a single amino acid substitution from Leucine^336^ to Tyrosine^336^ was indicated by a black arrow mark. (**c**) The sequence alignment shows the confirmation of mutations that were created within the infectious clones (pBP4-Δ1066-68) of CGMMV mutant (1–5 clones) and compared with the wild-type CGMMV infectious clone (pBP4). (**d**) The amplification of the pBP4-Δ1066-68 construct (16.4 kb) through the LC-PCR at different annealing temperatures. (**e**) The agro-infiltration of pBP4 constructs leading to the expression of typical green mottle mosaic symptoms in the systematic leaves of *N. benthamiana*, whereas (**f**) the agro-infiltration of pBP4-Δ1066-68 construct (clone-1) did not produce any symptoms in the infiltrated and systematic leaves of *N. benthamiana*. (**g**) The electron microscopy of the CGMMV mutant infected systematic leaves (inoculated with pBP4-Δ1066-68 construct) showed the presence of rod-shaped particles within asymptomatic tissues. (**h**) RT-PCR showing amplification CGMMV CP gene fragment (0.49 kb) using CP gene-specific primers (BM 556F and BM 557R). Green color: Wild type (CTG) and Red Color: Mutant (TAC).

**Table 1 viruses-15-02332-t001:** Details of the primers designed to manipulate the genome cucumber green mottle mosaic virus (CGMMV) through long-fragment circular-efficient PCR strategy.

Primer Name	Primer Sequences	Used for	Purpose
P1-F	CACCTCGAAAGCTTAG GGAGGGCAATCTTATGTTGA	Amplification of *NbPDS* gene from *N. benthamiana*	Development of pBP4-NbPDS constructs for silencing of *NbPDS* gene
P2-R	ATCAGAAGACCCTCGAA AGGAGGGTTACCATCTAAAA		
P3	1st PCR product as mega primer	Synthesis of pBP4 plasmid with the *NbPDS* sequence	
P4	TGCCCTCCCTAAGCTTTCGAGGTGGTAGCCTCTGACCAGACTACCG	Self-circularisation of pBP4 with *NbPDS* sequence	
P1a-F	GCTTCACAAGGTACCGCT ATGGCGAAGCGACCAGCAGAT	Amplification of capsid protein (CP) gene of ToLCNDV	Development of pBP4-∆CP-CPToLCNDV for the expression of ToLCNDV CP gene
P2b-R	ATTTGTGACCGAATCATAAAAATAG CACCATCAGAAGACCCTCGAAACTA		
P3c	1st PCR product as mega primer	Synthesis of pBP4-∆CP with the CP gene of ToLCNDV	
P4d	CGCTTCGCCATAGCGGTACCTTGTGAAGCAACTAGAAAATTAAG	Self-circularisation of pBP4-∆CP with the CP gene of ToLCNDV	
P5-R	CATCATCACCATCGACCCTAAACTG	Synthesis of full-length pBP4 from the deletion ends	Development of pBP4-∆Rep for the deletion of helicase and RNA-dependent RNA Polymerase gene from CGMMV genome
P6-F	GCTCTGGTTGACACAAGGATGCATTC		
P7-F	TCGATGGTGATGATGGCTCTGGTTGACACAAGGA	Self-circularisation of newly synthesized strands in pBP4 construct excluding the deleted Repportion	
P8-F	GTACGCAAAGTACTCAGATTTCGATTTC	Amplification of full-length BP4 with a terminal mutation at 1066th-68th nucleotide position	Creation of mutant CGMMV construct (pBP4-∆1066-68)
P9-R	AATACCCCTCCGATCTTCCAAGATAAAGCCAC		
P10-R	CGAAATCTGAGTACTTTGCGTACAATACCCCTCCGATC	Self-circularisation of pBP4 with mutation from CTG (leucine) to TAC (tyrosin)	

Red text indicates the flanking sequences of overlapping sequences; green text indicates the amino acid of silent mutation. ToLCNDV: Tomato leaf curl New Delhi virus.

**Table 2 viruses-15-02332-t002:** The success of genetic manipulation strategies of the cucumber green mottle mosaic virus infectious clone, pBP4 by long-fragment circular-efficient PCR.

Manipulation Strategy	Purpose	Size Manipulated (nt)	Total Size of Construct (Kb)	No. of Attempts	Average No. of Colonies	No Positive Colonies/ Tested by PCR	Success (%)
Insertion of *NbPDS* gene	Gene silencing	21 0	16.6	5	95	24/25	96
Insertion of CP_ToLCNDV_gene	Protein Expression	770	16.6	10	80	22/25	88
Deletion	Viral Replication	4400	12.0	3	150	25/25	100
Mutation	Phenotype	1	16.4	5	200	25/25	100

## Data Availability

Data are contained within the article and Appendix A.

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
