# Peer review of "Accelerated Long-Fragment Circular PCR for Genetic Manipulation of Plant Viruses in Unveiling Functional Genomics"

_viruses, 2023, doi:10.3390/v15122332_

Round 1
Reviewer 1 Report
Comments and Suggestions for Authors
Authors Jailani and coworkers presented here a manuscript entitled „Accelerated Long-Fragment Circular PCR for Genetic Manipulation of Plant Viruses in Unveiling Functional Genomics“.
The authors described the use of a modified PCR procedure to increase the possibilities of modification of plasmid vectors used to study the functional genomics of plant viruses. They also demonstrated the successful use of their constructs in the CGMMV-N. benthamiana model system.
The manuscript needs minor improvements and English corrections.
Figure 1 is illegible.
Reduce the number of self citations, especially 34-37.
Line 19: complimentary - should be complementary
Line 28: Different in planta assay – makes no sense
Line 34: better - ligation-free
Line 112: the sentence does not make sense
Comments on the Quality of English Language
Problems with the quality of the English language occur in the Introduction section, where the English language style is sometimes difficult to understand.
Some sentences are strangely constructed. Full stops are sometimes missing.
Author Response
Reviewer comment: The authors described the use of a modified PCR procedure to increase the possibilities of modification of plasmid vectors used to study the functional genomics of plant viruses. They also demonstrated the successful use of their constructs in the CGMMV-N. benthamiana model system.
Response: Thank you for your insightful comments. We appreciate your positive acknowledgment of our modified PCR procedure, which aims to enhance the flexibility in modifying plasmid vectors for investigating the functional genomics of plant viruses.
Reviewer comment: The manuscript needs minor improvements and English corrections.
Response: Thank you for your feedback. We appreciate your constructive comments. We have carefully addressed the identified areas for improvement and work on enhancing the English language throughout the manuscript.
Reviewer comment: Figure 1 is illegible.
Response: Thank you for bringing this to our attention. We apologize for any inconvenience caused by the illegibility of Figure 1. We have carefully reviewed the figure to ensure that it is presented clearly and make necessary adjustments in the revised manuscript.
Reviewer comment: Reduce the number of self citations, especially 34-37.
Response: Thank you for the suggestion. We will carefully review the self-citations in the specified range (34-37) and aim to reduce them in the revised manuscript. Our goal is to provide a balanced and relevant set of references.
Reviewer comment: Line 19: complimentary - should be complementary
Response: Corrected
Reviewer comment: Line 28: Different in planta assay – makes no sense
Response: Changed
Reviewer comment: Line 34: better - ligation-free
Response: Corrected
Reviewer comment: Line 112: the sentence does not make sense
Response: Changed
Reviewer 2 Report
Comments and Suggestions for Authors
This manuscript was very difficult to review due to the many inconsistencies/errors throughout the text. It also didn't help that the grammar was choppy and uneven with phrases stated that didn't really have any relevance to the topic being presented. While the LC-PCR methodology described in this manuscript is worthy of publishing, I'm left wondering that a similar result could be achieved much more simply by using something like the New England Biolabs (NEB) HiFi DNA Assembly approach since the authors are using the Phusion polymerase (from NEB) for amplifying the vector backbone.
I think the manuscript needs to be simplified to have a greater impact on the reader as I was reduced to going back and forth between the various sections of the manuscript to make sense of various statements. The primers, constructs, etc. are not uniformly mentioned or logically assigned which also leads to confusion. I truly appreciated the authors' supplying of a higher resolution Figure 1 that was actually legible and allowed me to understand the method in a much clearer fashion.
These are some of my main concerns:
1. It would've been extremely helpful to list all of the various constructs in one place (e.g. in Table 1 where the primers used are described and for what purpose). The construct name can be inserted under the Purpose column.
2. Ln. 133: is the construct pBP4-(delta)CP or is it pBP4-CPD (in Supplementary Figure 1)? Also, in the same figure, is it supposed to be pBP4 for the wild type CGMMV construct (figure caption) or is it pBMCG6.4 (figure legend)? Figures 2-4 all have the construct maps labeled as Bp4-etc. as opposed to the pBP4 throughout the body of the various sections.
3. Similarly, labeling of the other constructs is not consistently stated in the Materials and Methods section. e.g. constructs should be stated in Section 2.2 and not until the Results section for the various constructs.
4. I don't understand the rational for making the deltaRep mutant which is functionally dead. Unsurprisingly, leaves infiltrated with deltaRep did not produce replicating viral RNA. However, the fact that the plasmid construct had a dual 35S promoter capable of constitutive expression in the absence of viral replication leads me to question the sensitivity of the RT-PCR assay or the time frame dpi to detect RNA. I didn't notice the use of any viral RNA silencing suppressors for co-infiltration of Agrobacterium which may have led to less than optimal transcript stability.
5. I could not find a reference for why the particular amino acid point mutation was chosen. Was this based on a published report of this mutation producing no symptoms? If so, that needs to be cited. If not, what was the rationale for choosing that particular mutation?
6. Figures 2-4: I don't think there's a need to display the intermediate reaction products as the bands are typically very faint and rather meaningless in the grand scheme of things. Even if the bands were not detectable on a gel, doesn't mean that you won't get any mutants after transformation.
7. Figure 2, j and k: I am not a fan of semi-quantitative RT-PCR to show knock down of gene expression. It would be much better to use RT-qPCR.
8. Figure 3 is misleading as it shows the remnant CP size to be similar to the wt CP despite only 105 nucleotides remaining. The illustration needs to be altered to reflect a shorter CP region prior to the ToLCNDV CP. On the same note, what is the native size (in kD) of the ToLCNDV CP? Does the 29 kD band depicted in (i) a combination of the remnant CGMMV CP fused to the ToLCNDV CP? If not, how do you explain the cleavage of the fusion protein to yield the wt size CP?
Other minor corrections/comments:
a. Ln 116-117: these viral abbreviations need to be written as the full viral name as these have not been mentioned before (or anytime after for that matter).
b. Ln. 155: I did not see any mention of primer P4d? Is this supposed to be primer P4c by any chance?
c. Ln 192: it should be Wizard SV Gel and PCR... not Wizards.
d. Ln 199: what was the concentration of P4/P4c used?
e. Ln 200: isn't the total reaction volume 50 uL and not 25 uL as stated?
f. Ln 205: what is the concentration of DpnI used?
g. Ln 208: efficiency should be written with superscripts (105-106 cfu/ml)
h. Lns 211/230: Sanger sequencing should be capitalized as the technique was named after Frederick Sanger.
i. Lns 219, 224: primer concentrations?
j. Ln 265: the enzyme is Superscript, not Superscripts.
k. Ln 459: shouldn't it be LC-PCR instead of PC-PCR?
Comments on the Quality of English LanguageEnglish is disjointed and phraseology could be vastly improved. Need for proficient editing by English speaking/writing individual.
Author Response
Question 1: It would've been extremely helpful to list all of the various constructs in one place (e.g. in Table 1 where the primers used are described and for what purpose). The construct name can be inserted under the Purpose column.
Reply: Thank you very much. The names of the constructs developed through LC-PCR have been included in the purpose section of Table 1, also justifying the utility of different primers.
Question 2: Ln. 133: is the construct pBP4-(delta)CP or is it pBP4-CPD (in Supplementary Figure 1)? Also, in the same figure, is it supposed to be pBP4 for the wild type CGMMV construct (figure caption) or is it pBMCG6.4 (figure legend)? Figures 2-4 all have the construct maps labeled as Bp4-etc. as opposed to the pBP4 throughout the body of the various sections.
Reply: We have corrected and used the same name throughout the manuscript, including figures and supplementary material. The wild-type CGMMV construct was named as pBP4 (in vivo construct), derived from pBMCG6.4 (in vitro construct). The deletion constructs were named as follows: pBP4-∆CP (coat protein deleted), pBP4-∆Rep (replicase deleted), and pBP4-∆1066-1068 (trinucleotides at 1066-1068 mutated).
Question 3: Similarly, labeling of the other constructs is not consistently stated in the Materials and Methods section. e.g. constructs should be stated in Section 2.2 and not until the Results section for the various constructs.
Reply: Thank you for pointing out this issue. We have now corrected the names of the constructs and consistently stated them in all sections of the manuscript.
Question 4: I don't understand the rational for making the deltaRep mutant which is functionally dead. Unsurprisingly, leaves infiltrated with deltaRep did not produce replicating viral RNA. However, the fact that the plasmid construct had a dual 35S promoter capable of constitutive expression in the absence of viral replication leads me to question the sensitivity of the RT-PCR assay or the time frame dpi to detect RNA. I didn't notice the use of any viral RNA silencing suppressors for co-infiltration of Agrobacterium which may have led to less than optimal transcript stability.
Reply: The rationale for creating the ΔRep mutant (pBP4-ΔRep) was to demonstrate the LC-PCR protocol's capability to generate targeted deletion constructs. By majorly deleting the helicase and replicase domains, the RNA synthesized from the cDNA construct would be unable to self-replicate and move into systemic tissues. As a result, it would not be detected in systemic tissues. Importantly, no viral RNA silencing suppressors were co-infiltrated to ensure transcript stability.
Question 5. I could not find a reference for why the particular amino acid point mutation was chosen. Was this based on a published report of this mutation producing no symptoms? If so, that needs to be cited. If not, what was the rationale for choosing that particular mutation?
Reply: The mutations were randomly introduced in six different locations within the Rep (replicase) domain, guided by studies such as the one where mutations were created in the replicase region (https://apsjournals.apsnet.org/doi/pdf/10.1094/PHYTO-97-4-0412). However, this paper presents only one of these mutations. The details of the remaining mutations will be provided in a separate paper that will focus on the functional analysis of the asymptomatic infection construct. This study aligns with two other papers discussing the role of the replicase enzyme in the pathogenesis of other plant viruses.
Question 6. Figures 2-4: I don't think there's a need to display the intermediate reaction products as the bands are typically very faint and rather meaningless in the grand scheme of things. Even if the bands were not detectable on a gel, doesn't mean that you won't get any mutants after transformation.
Reply: Yes, it can be deleted, but keeping it would be helpful for researchers who intend to try this protocol. The pictures serve as a quick guide to avoid any confusion.
Question 7. Figure 2, j and k: I am not a fan of semi-quantitative RT-PCR to show knock down of gene expression. It would be much better to use RT-qPCR.
Reply: Unfortunately, we don't currently have live plant samples available for this assay. Consequently, it will be challenging to generate any RT-qPCR data immediately. Additionally, I would like to highlight that we have previously generated similar data for the knockdown of the NbPDS gene using another deletion-based construct (pBP4-∆Rep-NbPDS). The details of this study will be communicated soon in a separate manuscript.
Question 8. Figure 3 is misleading as it shows the remnant CP size to be similar to the wt CP despite only 105 nucleotides remaining. The illustration needs to be altered to reflect a shorter CP region prior to the ToLCNDV CP. On the same note, what is the native size (in kD) of the ToLCNDV CP? Does the 29 kD band depicted in (i) a combination of the remnant CGMMV CP fused to the ToLCNDV CP? If not, how do you explain the cleavage of the fusion protein to yield the wt size CP?
Reply: Thank you for bringing this to our attention. The original ToLCNDV CP size is 30.5 kD. We made a calculation mistake. According to the ToLCNDV CP amino acid size of 257, we should have calculated 25.7 kD. Additionally, the CGMMV partial retained CP size is 35 amino acids, which corresponds to 3.5 kD. Therefore, the total size is 29 kD (25.7 + 3.5 = 29 kD). Now, we have revised the calculation to consider the exact ToLCNDV CP size (30.5 kD) plus the CGMMV CP size (3.5 kD), resulting in a corrected total size of 34 kD. The required corrections have been implemented in Figure 3. We appreciate your understanding.
Other minor corrections/comments:
Question9: Ln 116-117: these viral abbreviations need to be written as the full viral name as these have not been mentioned before (or anytime after for that matter).
Reply: Corrected
Question10: Ln. 155: I did not see any mention of primer P4d? Is this supposed to be primer P4c by any chance?
Reply: We have made a correction in the primer list, changing "P4c" to "P4d." We apologize for the typo error and appreciate your understanding.
Question11: Ln 192: it should be Wizard SV Gel and PCR... not Wizards.
Reply: Corrected
Question 12: Ln 199: what was the concentration of P4/P4c used?
Reply: 1.0 micromoles
Question 13: Ln 200: isn't the total reaction volume 50 uL and not 25 uL as stated?
Reply: Corrected
Question 14:. Ln 205: what is the concentration of DpnI used?
Reply: 2U/µl, included
Question 15: Ln 208: efficiency should be written with superscripts (105-106 cfu/ml)
Reply: Corrected
Question 16: Lns 211/230: Sanger sequencing should be capitalized as the technique was named after Frederick Sanger
Reply: Corrected
Question 17:. Lns 219, 224: primer concentrations?
Reply: 1.0 micromoles
Question 18: Ln 265: the enzyme is Superscript, not Superscripts.
Reply: Corrected
Question 19:. Ln 469: shouldn't it be LC-PCR instead of PC-PCR?
Reply: Corrected
Round 2
Reviewer 2 Report
Comments and Suggestions for Authors
Much improved version.